# Graphene-Oxide-Based Fluoro- and Chromo-Genic Materials and Their Applications

**DOI:** 10.3390/molecules27062018

**Published:** 2022-03-21

**Authors:** Xiaoxiao Zheng, Rongli Zhai, Zihao Zhang, Baoqing Zhang, Jiangwei Liu, Aamir Razaq, Muhammad Ashfaq Ahmad, Rizwan Raza, Muhammad Saleem, Syed Rizwan, Syed Hassan Mujtaba Jafri, Hu Li, Raffaello Papadakis

**Affiliations:** 1Shandong Technology Centre of Nanodevices and Integration, School of Microelectronics, Shandong University, Jinan 250101, China; 202120353@mail.sdu.edu.cn (X.Z.); zhairls@163.com (R.Z.); zhangzihao987@163.com (Z.Z.); 201812324@mail.sdu.edu.cn (B.Z.); 2School of Energy and Power Engineering, Shandong University, Jinan 250061, China; jiangwei.liu@sdu.edu.cn; 3Department of Physics, COMSATS University Islamabad, Lahore Campus, Lahore 54000, Pakistan; aamirrazaq@cuilahore.edu.pk (A.R.); maahmad@cuilahore.edu.pk (M.A.A.); rizwanraza@cuilahore.edu.pk (R.R.); 4Institute of Physics, The Islamia University of Bahawalpur, Bahawalpur 63100, Pakistan; msaleem@iub.edu.pk; 5Department of Physics, National University of Sciences and Technology, Islamabad 44000, Pakistan; syedrizwan@sns.nust.edu.pk; 6Department of Electrical Engineering, Mirpur University of Science and Technology (MUST), Mirpur 10250, Azad Jammu and Kashmir, Pakistan; hassan.jafri@must.edu.pk; 7Department of Materials Science and Engineering, Uppsala University, 75121 Uppsala, Sweden; 8Department of Chemistry, Uppsala University, 75120 Uppsala, Sweden; 9TdB Labs AB, Uppsala Business Park, 75450 Uppsala, Sweden

**Keywords:** composite materials, graphene oxide, dyes, pigments, inks, fluorogenic applications, sensors

## Abstract

Composite materials and their applications constitute a hot field of research nowadays due to the fact that they comprise a combination of the unique properties of each component of which they consist. Very often, they exhibit better performance and properties compared to their combined building blocks. Graphene oxide (GO), as the most widely used derivative of graphene, has attracted widespread attention because of its excellent properties. Abundant oxygen-containing functional groups on GO can provide various reactive sites for chemical modification or functionalization of GO, which in turn can be used to develop novel GO-based composites. This review outlines the most recent advances in the field of novel dyes and pigments encompassing GO as a key ingredient or as an important cofactor. The interactions of graphene with other materials/compounds are highlighted. The special structure and unique properties of GO have a great effect on the performance of fabricated hybrid dyes and pigments by enhancing the color performance of dyes, the anticorrosion properties of pigments, the viscosity and rheology of inks, etc., which further expands the applications of dyes and pigments in dyeing, optical elements, solar-thermal energy storage, sensing, coatings, and microelectronics devices. Finally, challenges in the current development as well as the future prospects of GO-based dyes and pigments are also discussed. This review provides a reference for the further exploration of novel dyes and pigments.

## 1. Introduction

Dyes and pigments play an important role in our daily life because they can be widely used in many applications as key ingredients in cosmetics, paints, textiles, coatings, plastics, construction materials, food, paper, and printing inks [1]. The global colorants market has in recent years been greatly enhanced as a result of the steadily increasing needs of various end-use industries pertaining to dyes and pigments. More and more synthetic dyes and pigments are being produced. However, the applications of some synthetic dyes and pigments have detrimental effects on the environment [2]. With the development of science and technology, some traditional dyes and pigments can no longer meet the growing demand for color performance, corrosion resistance [3], photoelectric conversion efficiency [4,5,6], etc. For example, some organic dyes such as phthalocyanines tend to form aggregates due to their unique structure, which adversely influences their color performance in terms of color dispersion and strength. Conversely, pigments can improve the corrosion resistance of waterborne epoxy resins. However, some inorganic anticorrosive pigments have a serious impact on the environment and some traditional pigment-based epoxies cannot be used in harsh environments. In order to overcome these limitations and further improve the performance of traditional dyes and pigments, some modification and/or functionalization is necessary. Among the surface modifiers, materials of the graphene family have attracted considerable attention, being used as scaffolds or as important cofactors influencing the properties and performance of the composite materials in which they are employed [7].

Graphene is an outstanding representative of the 2D materials family and it is also the most studied 2D material, exhibiting a range of remarkable properties such as outstanding mechanical strength, excellent electrical conductivity, high thermal conductivity, good light transmittance, ultrahigh carrier mobility, large surface area and flexibility [8,9,10,11,12,13,14,15]. Furthermore, graphene shows complete impermeability to gases. These properties make graphene have a very wide range of applications in flexible electronics, protective coatings, solar cells, etc. [15,16,17,18,19,20,21]. However, graphene is a hydrophobic material and it has no noticeable solubility in molecular solvents, which greatly limits its application. Graphene oxide (GO), as a hydrophilic derivative of graphene, has excellent dispersibility in many solvents, which is ascribed to the presence of a large amount of oxygen-containing functional groups on the surface of GO. These functional groups also provide reactive sites for surface modification reactions, thus opening up numerous possibilities for the modification of dyes and pigments. The composite dyes/pigments may show improved performance due to the interaction between functional molecules. For example, Guo et al. [1] prepared GO-modified polymer dyes by introducing GO into aqueous polyurethane-based polymer dyes. The modified polymer dyes exhibited high coloration rates and their dry/wet rubbing fastness and migration resistance were both improved to grade 5, which promoted their application in high-performance leather dyeing. Sadawy et al. [22] synthesized a GO/zinc phosphate composite coating (GO/ZP) on an Al-Zn-Mg alloy substrate using a chemical conversion coating technique, proving that the addition of GO to a ZP bath can improve the corrosion resistance of the coating. Due to the special structure of GO, it shows great potential in the fabrication of novel dyes and pigments. In this work, a detailed review of graphene-based dyes and pigments is presented. The recent progress of GO-based dyes and pigments, especially the effects of GO on the different performances of the synthetic dyes/pigments and the applications of GO-based dyes and pigments, are highlighted.

## 2. Properties of GO

GO is a unique two-dimensional material with a large number of oxygen-containing functional groups on its surface and edges, such as hydroxyl, carboxyl, and epoxy [23]. The large specific surface area and the introduction of oxygenated functional groups enable GO to be stable in water and organic solvents [24,25], and also provide many chemical reaction sites for the synthesis of GO-based functionalized composites [26]. GO exhibits excellent electronic, optical, thermal, and mechanical properties, and chemical activity, making it a good candidate material for many applications [27], such as polymer composites, energy storage, sensors [28,29], field effect transistors (FETs), biomedicine [30,31], transparent conductors for organic photovoltaic cells [32], and electrodes for electrochemical applications [33]. Here we will briefly introduce the electronic and optical properties and chemical reactivity of GO.

### 2.1. Electronic Properties

As shown in Figure 1, GO has a two-dimensional honeycomb lattice structure similar to graphene and there are many oxygen-containing functional groups on its surface [34,35]. Due to the existence of oxygen-containing functional groups, the synthesized GO is usually insulating, and the resistance value of monolayer GO is about 10^12^ Ω/sq [36]. The electrical conductivity of GO flakes depends on the degree of reduction, which is related to the ratio of the graphite region (sp^2^) to the oxidized region (sp^3^). The corresponding result is shown in Figure 2 [37].

Based on first principles, Yan et al. [38] studied the arrangement of oxygen-containing functional groups on graphene and further investigated how they affect the electronic properties of GO. They found that when epoxy groups and hydroxyl groups are located on both sides of graphene, the energy is significantly reduced, indicating that they are more likely to aggregate on the surface of graphene. They also obtained a local density approximation (LDA) band gap ranging from a few fractions of eV to 4 eV by varying the degree of oxidation and the position of oxygenated functional groups. In addition, a large number of studies have shown that the conductivity and mobility of reduced GO (rGO) are lower than those of pristine graphene [8]. There are two main reasons for this: (1) the unreduced functional groups disconnect the π- delocalized orbital network, thus limiting the charge penetration; (2) the trap state generated during the reduction process prevents the migration of carriers.

### 2.2. Optical Properties

GO has adequate optical properties such as tunable optical transmittance, fluorescence, and nonlinear optical properties [39].

#### 2.2.1. Tunable Optical Transmittance

GO has a high optical transmittance in the visible region [40]. It has been reported that the transmittance of GO is related to its thickness and oxygen content [39]. As shown in Figure 3, when the thickness of GO increases from 6 nm to 41 nm, its transmittance in the visible wavelength range decreases from 90% to 20%, indicating that the transmittance of GO decreases with increasing thickness [36]. The transmittance also decreases with the increase in the degree of oxidation. As illustrated in Figure 4, the transmittance of chemically reduced and thermally annealed GO thin films decreases from 95% to 60% at the wavelength of 550 nm [41]. The transparency of GO makes it more suitable for use in the display industry [42].

#### 2.2.2. Fluorescence

The introduction of oxygenated functional groups opens the optical band gap of GO and generates fluorescence. By adjusting the ratio of the graphite/graphene region and the oxidation region to change the electronic energy band structure, fluorescence can be generated in the ultraviolet, visible, and near-infrared regions. Based on this property, GO can be applied in the field of light-emitting diodes and optoelectronics [28,29,30,33,40].

The fluorescence in GO arises from the recombination of electron–hole pairs in various localized electronic states. As shown in Figure 5, the chemically modified GO shows three photoluminescence (PL) bands, which are generated by specific electronic transitions between different orbitals, including σ* to n transition, π* to π transition and π* to n transition [43]. Among the above three types of transitions, the first transition has the largest energy gap, while the π* to n transition that occurs between the anti-bonding and the bonding molecular orbitals has the smallest energy gap in the long-excitation-wavelength region. In addition, the features of the PL emission of GO will have an impact on chemical sensing, biosensing, and photovoltaics [29].

#### 2.2.3. Nonlinear Optical Properties

Two different nonlinear optics (NLO) regions (sp^2^ region and sp^3^ region) of ultrafast optical dynamics are shown in Figure 6 [44]. The NLO properties of GO are determined by the combined effect of sp^2^ and sp^3^ regions. In the sp^2^ region, the NLO absorption is mainly saturated absorption, while in the sp^3^ region, it is two-photon absorption. Through the controlled reduction process, the two-photon absorption gradually transforms to saturable absorption, which can be used for shorter detection wavelengths. Therefore, the NLO response can be tuned by adjusting the oxidation degree of GO and the position of the functional group. This feature gives GO a potential application in optoelectronic devices.

### 2.3. Chemical Reactivity

Oxygen-containing functional groups on the surface and edges of GO make it more chemically active than pristine graphene. The most important chemical reaction of GO is reduction, and GO can be reduced to rGO by various methods. It has been reported that GO can be reduced by heat treatment or electrochemical treatment using strong reducing agents such as hydrazine [45] and sodium borohydride [46]. The resulting product is very similar to the pristine graphene and has been widely used in physical and engineering applications. In addition, a wide range of chemical functionalizations of GO are known. GO is an important material for the preparation of functionalized-graphene-based composites. Other groups can be added to GO by various covalent and non-covalent connections. For instance, small molecules as well as polymers can be covalently attached to the carboxyl [47], epoxy [48], and hydroxyl groups of GO, or non-covalently attached to the surface of chemically modified graphene (CMG) [49,50,51]. The chemical activity of GO promotes its application in polymer composites, sensors, and optoelectronics.

## 3. Categories of GO-Based Materials Involving Dyes and Pigments

Due to the large amount of oxygen-containing functional groups (including epoxy and hydroxyl groups on sp^3^ hybridized carbon on the basal plane, as well as the carbonyl and carboxyl groups located at the sheet edges on sp^2^ hybridized carbon) [39,52,53,54], GO has been widely used as the support to fabricate novel dyes and pigments. These polar oxygen-containing functional groups of GO make it highly hydrophilic. This allows GO to have excellent dispersibility in many solvents, especially in water [55]. This is very important for further processing and derivatization. In addition, the oxygen-containing functional groups can provide multiple reactive sites for chemical modification or functionalization of GO, which is beneficial for developing GO-based materials. The surface functionalization can be broadly divided into the following two methods: covalent functionalization and non-covalent functionalization [8,18,56]. These oxygenated groups have greatly expanded the application scopes of graphene. The obtained hybrids not only combine the specific chemical and optical properties of each component, but also possess the ability to introduce some new properties. As a result of the unique structures and properties, GO-based dyes and pigments are a contemporary research frontier and show great potential in diverse fields such as coatings, supercapacitors, photocatalysis, energy-storage materials, and so on. In this section, we review the role of GO in the construction of new dyes and pigments as well as the application of GO in new ink technology.

### 3.1. GO-Based Materials Involving Dyes

Regulation of dye aggregation has always been a research hotspot in the field of colloids. Many dye molecules are inclined to form dimers or aggregates due to the existence of multifarious intermolecular non-covalent interactions, such as hydrogen bonds, van der Waals interactions, π–π stacking, and electrostatic interactions. Under different forces, the dye molecules can be arranged in diverse forms, resulting in dyes having special photochemical and photophysical properties. Song et al. [57] developed a facile method to fabricate GO-dye composite film by using the Langmuir–Blodgett (LB) technique. GO sheets with many functional groups and large specific surface area play a key role in the aggregation of organic dye molecules (congo red, methylene blue, and rhodamine B). They can induce dye molecules to self-assemble and form highly ordered GO-dye LB film containing H-aggregates and/or J-aggregates by π–π interaction as well as electrostatic interaction between the negatively charged GO and cationic dye molecules. The LB films show huge application potential in biological fields, optical elements, and so on.

GO can not only induce dye molecules to form various aggregates, but it also can reduce the aggregation degree of dyes. Sun et al. [58] grafted rhodamine B hydrazide (RBH) dye onto GO via covalent bonds, preparing RBH functionalized GO (RBH-GO). The addition of GO greatly reduces the aggregation degree of RBH by deteriorating π–π stacking. In other words, the distribution of the RBH electron cloud can be changed when covalently grafted onto GO. The optimized geometry of RBH before and after covalent grafting onto GO is shown in Figure 7. Therefore, the obtained product exhibits excellent monodispersion. Furthermore, since the surface of GO possesses a large number of polar oxygen-containing functional groups, the hydrophilicity of RBH-GO can be significantly improved compared with the single RBH.

In addition to changing the aggregation degree of dyes, covalent functionalization of dyes with GO can further change the properties of dyes. When the organic chromophores in dyes are covalently attached to GO, their aromatic properties will be disturbed, making it possible to modulate their optoelectronic properties. Chen et al. [59] designed and prepared GO ternary nanohybrids co-functionalized by phenyl porphyrins and thieyl-appended porphyrins (TPP-GO-TTP) for the application of optical limiting devices. The synthetic route of TPP-GO-TTP is shown in Figure 8. The nanohybrids exhibit better nonlinear absorption and optical limiting performance compared to their components (Figure 9). This is due to the effective charge transfer effect between the porphyrins (electron donors) and GO (electron acceptors).

TPP-GO-TTP is a kind of donor-acceptor nanohybrid. This nanohybrid is currently a very important research direction and it has attracted widespread attention because the covalent linkages between electron donors and electron acceptors can introduce some novel properties rather than a simple combination of their individual properties. Besides the porphyrins, phthalocyanine dye is another important electron donor. Wang et al. [60] prepared a new donor-acceptor-type nanohybrid material. The configuration of the nanohybrid is illustrated in Figure 10. The tetrakis(4-aminophenyl)porphyrin (TPP1) and tetraaminophthalocyanine (ZnPc) are covalently bonded to the GO through amido bonds. The as-prepared TPP1-GO-ZnPc not only possesses the intrinsic characteristics of porphyrin, phthalocyanine, and GO, but also has some new properties arising from the mutual π interactions between porphyrin, phthalocyanine, and GO. In comparison with GO, TPP, ZnPc, TPP-GO, and ZnPc-GO, TPP1-GO-ZnPc exhibits enhanced optical limiting performance (Figure 11), which can be explained by the more effective charge transfer between the electron donor (TPP1 and ZnPc) and electron acceptor (GO). Furthermore, GO with a large specific area can load more organic dyes, which also plays an important role in improving the optical limiting.

GO plays a very important role in the construction of new dyes. This is attributed to the excellent platform of GO, which can provide many reactive sites for organic functional dyes, greatly improving the performance of traditional dyes. The as-prepared dyes have been widely used in many fields owing to their unique characteristics. In addition to the field of dyeing and photoelectric devices, they also have extensive applications in solar-thermal energy storage. Some organic pigments and dyes with conjugated structures can absorb and convert light energy to thermal energy, such as anthraquinone dyes, polyaniline, and azo dyes [61,62]. Among these, anthraquinone dye has high molar extinction coefficients and the ability of strong visible light selective absorption, showing outstanding performance in the field of photothermal conversion. Nevertheless, the fluorescence generated by radiative leapfrogging can cut down its light-to-heat conversion efficiency to a certain extent. In order to improve the photothermal conversion efficiency, Wang et al. [63] grafted blue anthraquinone dyes (Bdye) onto GO (GO-co-Bdye) through amide reactions between the carboxy groups of GO and the amino groups of the dye, synthesizing a novel GO-based dye. GO, acting as a fluorescent receptor, can receive the fluorescence generated by the dye and convert it to thermal energy, which greatly improves the photothermal conversion performance of GO-co-Bdye. Via the method of blending and impregnating of the modified dyes and poly(ethylene glycol) (PEG), a new phase change material (PCM) can be obtained, which can be used in photo-driven energy conversion and storage. The preparing processes and mechanism are illustrated in Figure 12. The introduction of GO can not only enhance the photothermal effect, but can also serve as a supporting-material for PCM, improving the thermal stability, crystalline nature, and thermal conductivity of PCM. Therefore, GO plays an important role in the synthesis of efficient solar energy conversion materials.

The existence of oxygen-containing functional groups on GO means it is often used as the starting material for fabrication of graphene-based dyes by acylation reactions. As well as the usual properties of GO, the property of no obvious toxic effects in vivo has also attracted the attention of researchers. If the fluorescent dyes are covalently functionalized onto GO, the application of dyes in bioimaging can be greatly expanded. Aminocoumarin dyes, which have high extinction coefficients and large Stokes shifts, can fluoresce in the blue-green spectral region. They can be covalently functionalized with GO to obtain a novel fluorescent nanohybrid (GO-NH-COUR) [64]. As shown in Figure 13, the 4-methyl-7-aminocoumarin (NH_2_-COUR) is grafted onto GO by the formation of amide bonds with GO. Compared with the fluorescence spectra of NH_2_-COUR, a slight bathochromic shift can be observed in the fluorescence spectra of GO-NH-COUR, which is ascribed to the formation of new luminescent centers at the surface of GO-NH-COUR. In addition, aminocoumarin plays an important role in the fluorescence property of the prepared nanohybrid. Due to the interaction between GO and NH_2_-COUR, GO-NH-COUR possesses excellent dispersion in aqueous solution and good optical properties. It can not only be used to reveal how the vis-NIR fluorescence of GO-based dye responds to pH, but can also be transfected into living cells for fluorescence imaging. Therefore, combining dyes and GO to prepare nanocomposites that can absorb and emit light in visible spectral regions will be an important research direction.

Due to its chemical structure and unique performance, GO can serve as an ideal nanoscale building block for graphene-based hybrid materials. The addition of GO has a huge effect on the properties of graphene-based hybrid materials. For example, the addition of GO can improve the color performance of dye by regulating the aggregation degree of dyes. GO can also modulate the optoelectronic properties of some dyes through the mutual π interactions between GO and dyes. With the development of materials science, more and more reports about graphene-based dyes are anticipated and a great application potential in dyeing, photoelectronic devices, and sensing is envisioned.

#### GO and rGO as Platforms for Fluorogenic Sensing and Relating Applications

In recent years, it has been clearly shown that GO is a perfect platform capable of complexing to a range of fluorogenic dyes causing the quenching of their fluorescence in a very efficient and controlled fashion. Due to this, GO has found applicability in a wide range of scientific and technological areas involving fluoreogenic sensing. Applications spanning from sensors of metal or ametal ions to gases or biologically relevant molecules, and finally to photocatalysis and phototherapy, are well known.

Prominent examples of this type of material are the complexes of GO with fluorescein. Fluorescein is a protype fluorescent compound with numerous derivatives that are widely used as labelling agents. When fluorescein is combined with GO or rGO, various opportunities are generated such as sensing of specific ions or gases. Sharma et al. [65] recently developed such a system involving fluorescein that is useful for the detection of As(III) in drinking water. Nanoparticles of a complex between rGO and fluorescein were found to exhibit an As(III)-concentration-dependent fluorescence enabling the determination of As(III) ions at concentrations as low as 0.96 µg/L.

In a similar fashion, Hunag et al. [66] reported on a dye sensor assay based also on fluorescein and rGO yet with a different application scope. Huang et al. employed the quenching of fluorescence due to the interaction of rGO and fluorescein and the recovery of fluorescence, which occurs when an antagonist dye is adsorbed by rGO, displacing fluorescein. The extent of displacement and recovery of fluorescence of fluorescein is proportional to the concentration of the target dye and therefore this structurally simple assay can find application in the detection and quantitative analysis of a variety of dyes. Wang et al. [67] recently developed a fluorescence biosensing approach that involves GO and fluorescein complexes. The same strategy utilizing the strong quenching of fluorescence induced by the complexation of fluorescein to GO was followed by this research group for the interaction of fluorescein with targeted DNA (analyte), which led to recovery of fluorescence in an analyte-concentration-dependent fashion.

Cyanine dyes are also of high relevance in diagnostics, fluorescence imaging, and intravital microscopy. GO is known to drastically affect the fluorescence of cyanine dyes and this is why they are combined with GO for a range of applications involving interesting fluorescence phenomena [68]. Youn et al. [69] recently developed various multiplexed sensing platforms consisting of GO and different cyanine dyes. By employing the fluorescence resonance energy transfer (FRET) methodology similar to that described before (vide supra), they achieved sensing of a range of antibiotics (such as sulfadimethoxine, kanamycin, and ampicillin) with a sensing capacity of very low concentrations of antibiotics, even as low as nearly 2 ng/mL. Guo et al. [70] exploited the pH-responsive (and generally stimuli-responsive) character of a cyanine dye (cypote), which they grafted onto GO, achieving a GO-based nanoplatform in which the conformation variability at different pH values in aqueous solutions can trigger pH-responsive photothermal effects. These effects were not employed in a sensing system but as a useful tool causing severe cell damage, which could in turn be applicable in tumor ablation. The research field of GO- and rGO-based fluorogenic sensing applications is steadily growing and there is an undoubtedly wide range of analytes that can be targeted and analyzed with extremely high accuracy.

### 3.2. GO-Based Pigments

Pigments comprise natural coloring matter, usually in powder form, and they are ordinarily insoluble in water and some other organic solvents [71,72]. These materials are an important part of paints and coatings in the world, and they are also the most important ingredients in water-based coatings and ink formulations [73,74,75,76,77]. With the continuous development of various industries, the demand for pigments continues to increase [72,78]. Therefore, in order to fabricate pigments with excellent performance, great research interest has been attracted by the modification of initial pigments or the design of novel pigments with GO. According to their application, morphology, structure, and other characteristics, the as-prepared pigments can be divided into different categories.

#### 3.2.1. Colorful Pigments

As is well-known, the particle size and distribution are essential parameters that can affect pigment performance (e.g., shade, flow behavior, tinctorial strength) [79,80]. Complete dispersion of pigments in solvents will ensure the cleanliness of shade, optimum tinctorial strength, and good gloss in the final products. Surface modification is the usual method to improve the color performance of organic pigments. Among various surface modifiers, such as anionic, cationic, and nonionic surfactants; hyperdispersants, abietic acid, organic amines, and synthetic polymers; GO has attracted widespread attention owing to its special properties and structures. Lv et al. [79] used GO to modify two copper phthalocyanine blue pigments, C.I. Pigment Blue 15 (B) and C.I. Pigment Blue 15:3 (BGS), obtaining two modified pigments. The addition of GO not only has no effect on the pristine chemical structures of pigments, but can also reduce the aggregation effectively and result in more uniform particle size distributions. In addition, the modification of pigments also improves the color strength. Due to the abundant hydroxyl and carboxylic acid groups in GO, the wettability of the original pigments is greatly improved.

GO can not only improve the initial performance of pigments by modification of pigments, but can also be used to synthesize eco-friendly and non-fading pigments. To prepare this kind of pigment, amorphous photonic structures (APSs) have attracted researchers’ attention. Among APSs, the short-range-ordered APS plays an important role, which is due to the fact that it can produce angle-independent non-iridescent structural colors when the structural characteristic sizes are comparable to the visible wavelengths [81,82]. Song et al. [83] used GO as a functional additive to mix with monodispersed polystyrene (PS) spheres, synthesizing non-iridescent structural colors. On account of the oxygen-containing functional groups on the surface of GO, GO can disperse well with a PS sphere. The PS sphere is trapped and encapsulated in the GO network structure, resulting in hindered equilibrium-state transmission and the crystallization of particles. As a result, GO and PS self-assemble into an amorphous arrangement and show non-iridescent structural color. After reduction treatment of the mixture using hydrated hydrazine, the contrast and visibility of structural color can be significantly improved due to the absorption of the multistep scattered lights by reduced GO. What is most important is that the color spectral purity and the hue can be tuned by changing the concentration of GO and the diameter of the PS sphere. The corresponding color cards are shown in Figure 14.

#### 3.2.2. Anticorrosive Pigments

Pigments are not only rich in colors, but they are also one of the basic ingredients of primer formulations. They can be used as inhibitors to decrease the corrosion of metallic substrate surfaces. In order to fabricate environmentally friendly and nontoxic corrosion inhibition pigments, some novel nanomaterials have attracted much attention, especially nanomaterials based on GO. For example, Xue et al. [84] used GO as a precursor to fabricate an efficient anticorrosion pigment via in situ bonding technology. The presence of oxygen-containing functional groups (–COOH) on GO can provide active sites for in situ growth of hydroxyapatite (HAP), forming GO-hydroxyapatite (GO-HAP) nanocomposites. The GO-HAP exhibits excellent anticorrosion performance owing to the synergistic protection of GO and HAP. The anticorrosion mechanism of GO-HAP is depicted in Figure 15. GO sheets can collect electrons released from the anode, inhibiting the cathodic reaction. HAP can immobilize the aggressive Cl¯ and generate stable chlorapatite. When the GO-HAP is added into the epoxy resin system, a tightly structured anticorrosive coating can be obtained, which is ascribed to the GO playing the role of an adhesive between the pigment and epoxy resin. Compared with the blank epoxy resin system, the impedance value of 0.6% GO-HAP/epoxy increases by 754.4%. This result shows that GO-HAP pigment has an excellent anticorrosive property, which can protect the steel matrix from corrosion.

Additionally, abundant oxygen-containing groups on the surface and edge of GO can provide potential reactive sites for polymer-grafted modification [53,54,85,86,87,88]. Polymer-grafted GOs have gained extensive attention in recent years due to their extraordinary advantages. In particularly, they have been widely reported as anticorrosive pigments in waterborne epoxy coatings by forming a layered barrier and increasing the diffusion pathway of corrosive electrolytes. Zhu et al. [89] synthesized a compound pigment of polypyrrole-functionalized GO (GO-PPy) by an in situ polymerization method. Incorporation of the nanocomposite into waterborne epoxy coating can obviously improve the corrosion protection performance of the coating. The composite coating was painted on the surface of mild steel by wire rod brushing and the thickness of the coating layer was about 80 µm. The mechanism of corrosion resistance of the GO-PPy coated on mild steel is shown in Figure 16. The GO nanosheet perpendicular to the thickness of the coating provides a meandering diffusion channel for corrosive ions, which delays the occurrence of corrosion. In addition, the amino functional groups of PPy generate a crosslinking effect to the epoxy, decreasing the micro-pores in the epoxy. The conducting PPy can accept electrons and accelerate the formation of a passivation layer composed of Fe_3_O_4_ and Fe_2_O_3_.

The severe economic losses caused by metal corrosion and increasing environmental pollution have stimulated researchers to explore more efficient, environmentally friendly, and non-toxic anticorrosive pigments. One of the attractive methods for improving the anticorrosion performance of pigments is to fabricate a composite with nanomaterials. Among various nanomaterials, GO has attracted widespread attention due to its excellent properties. For example, the lamellar structures of GO are good barriers to oxygen, corrosive ions, and water and the electron collection ability of GO can inhibit the electron transfer during the corrosion reaction. The presence of oxygen-containing functional groups on GO enables it to be covalently functionalized with many traditional pigments, further improving the anticorrosion performance due to the synergistic effect. With the advancement of technology, it is believed that GO will make a greater contribution in anticorrosion pigments.

### 3.3. New Ink Technologies

The development of ink has a long history, which can be traced back to China in the third century AD [90]. The inks used at that time were mixed from lamp-black (dark pigment) and gum (binder) dissolved in water. With the development of printing processes, oil-based inks appeared and began to develop rapidly. The development of printing technology has stimulated the development of the ink industry. Therefore, fabricating inks with multiple functions and high quality has become a research hotspot.

Graphene, as one of the potential 2D materials, offers an ideal platform for next-generation disruptive technology [91,92]. In recent years, with the development of new ink formulations related to effective printing technologies (including screen printing, gravure printing, spray coating, inkjet printing, and extrusion-based printing), interest in the use of graphene and its derivatives to make functional inks has grown exponentially [90]. The introduction of 2D materials such as graphene has greatly promoted the development of ink formulations, allowing the printing to be carried out on various substrates such as plastics and metal substrates. In 2011, Huang et al. [93] first reported a graphene-based conductive ink for the direct injection printing of high-image-quality patterns on plastic substrates and applied this simple technique to fabricate flexible electronic circuits and an electrochemical sensor. Torrisi et al. [94] used graphene-based ink to print thin-film transistors through inkjet printing in 2012, which paved the way for all-printed graphene devices on arbitrary substrates. Since then, this research field has experienced a rapid improvement in ink formulation methods and equipment manufacturing.

#### 3.3.1. Basic Ink Composition

A basic ink formulation usually includes pigments, binders, solvents, and various additives [95]. Among these, the binders and solvents, also known as ink varnishes, form liquid carriers of other ink components. The choice of ink varnishes lies in the specific printing technologies, substrates, and end applications. Pigments, as one of the important components in ink, have experienced many revolutions with the development of materials science. Over the past twenty years, more and more attention has been paid to adding functional materials (such as conductive materials, semiconductors, dielectric materials) as active pigments to inks because these materials can lend their characteristics to prints, allowing the development of various applications [16,96,97]. The effect of additives is to change the properties of inks and the printed film. Although they are added in a small proportion, they have a significant impact on the performance of inks and the final characteristics of the dry film.

#### 3.3.2. Graphene-Based Inks

Graphene and its derivatives each have their own unique advantages. Introducing them into the printing ink system to replace the pigment component can enable preparation of graphene-based ink [10,98]. This 2D ink system is different from traditional ink, because it is designed to have some certain functions rather than the graphical property. In 2D ink formulation, some characteristics such as color strength and gloss are usually not important considerations. Ink viscosity and rheology are two important factors that should be taken into account in the initial design of graphene-based ink. These two key parameters will play a significance role in the printability of an ink. According to the difference in viscosity, inks can be roughly divided into two categories: paste inks and liquid inks. Paste inks are colloidal suspensions with high viscosity. They exhibit non-Newtonian fluid behavior, which means that shear force must be applied to make the ink flow before printing. Screen printing and letterpress printing demand high-viscosity ink. In order to prepare an ink with a suitable viscosity, adding polymeric binders is usually necessary. For example, Loh et al. [99] changed the viscosity of ink by adding ethyl cellulose, thereby successfully preparing a graphene-based ink that can be used for direct ink writing. Compared with paste inks, the viscosity of liquid inks is naturally far lower and inkjet printing, gravure printing, and flexographic printing usually need liquid inks.

Different printing technologies have different requirements for ink viscosity. Reasonable control of ink viscosity is very important to obtain high-performance inks and they show great potential applications in flexible, thin, and wearable electronics. For example, Lacey et al. [100] printed complex hierarchical porous structures using extrusion-based printing technology, which were proved as the first 3D-printed Li-O_2_ cathodes. The additive-free and printable ink was produced by adding the holey GO (hGO) with high concentration (about 100 mg mL^−1^) to water. The aqueous hGO ink exhibits shear-shinning behavior, and can be extruded into complex 3D architectures (such as stacked mesh structures) by extrusion-based printing. The printed hGO mesh has multiple levels of porosity from macroscale to nanoscale, which provides pathways for electrolytes and oxygen, improving the performance of Li-O_2_ batteries. Gonzalez-Dominguez et al. [101] prepared some water-based inks composed of a ternary system (carbon nanotubes, GO, and nanocellulose) by autoclave methods. Importantly, by controlling the experimental conditions, low-viscosity inks, high-viscosity paste, or self-standing hydrogels can be obtained. The obtained liquid inks as well as the viscous pastes can be printed into conductive films with low resistivity value (less than 100 Ω/□) by spray coating and rod-coating technologies, respectively. These conductive films possess excellent robustness, which has the ability to avoid decomposition during corrosive treatment with organic solvents, and their electrical properties can be improved through high-temperature treatment, showing great potential in various applications such as batteries and solar cells. Chang et al. [102] prepared a water-soluble graphene@polyvinyl alcohol (PVA)-H_3_PO_4_ hybrid ink for the fabrication of microelectrodes of planar supercapacitors by the gravure printing technique. The high dispersibility of graphene in PVA-H_3_PO_4_ effectively prevents the self-restacking/aggregations of graphene, achieving the enhanced accessibility of electrolyte ions to the graphene surface. The flexible supercapacitors show enhanced electrochemical performance. The increased areal capacitance is 37.5 mF cm^−2^ at the scan rate of 5 mV s^−1^ and the maximum energy density of 5.20 μWh cm^−2^ is obtained at the areal power density of 3.2 mW cm^−2^. Wang et al. [103] reported a polyaniline/GO (PANI/GO) gel ink for three-dimensional (3D) printing. The addition of GO can effectively adjust the rheological properties of PANI, making the viscosity of composite ink meet the requirements of the direct ink writing technique. In addition, the reduction of GO can further strengthen the conducting mechanical strength of PANI. The printed PANI/RGO interdigital electrodes of planar supercapacitors exhibit satisfactory electrochemical performance with an areal-specific capacitance of 1329 mF cm^−2^.

Recently, functional graphene-based inks have developed rapidly and they show great potential application in the construction of portable and flexible electronic devices including supercapacitors, sensors, batteries, solar cells, and optoelectronics. On the basis of the requirements of different printing techniques, regulation of the formulation of ink is a crucial subject, because it plays an important role in the performance of printed devices. At present, more and more novel materials with excellent properties are being applied into the preparation of composite inks. The exploration of functional inks further promotes the development of electronic devices towards the direction of miniaturization, flexibility, and versatility.

## 4. Conclusions

GO is currently being used extensively in the fabrication of novel dyes and pigments owing to its excellent properties and special structure. Combining GO and/or other members of the graphene family with classic dyes and pigments has a great impact on the performance of the newly arising composite materials. For example, GO can effectively regulate the aggregation of dyes, modulate the original optoelectronic properties of dyes, improve the color strength of pigments, enhance the anticorrosion performance of pigments, and lead to the fabrication of functional inks. The fabricated composites show remarkable applicability in various technological and scientific areas within biology and fluorescence imaging, optical elements, solar-thermal energy storage, sensing, coatings, and microelectronic devices. Although GO-based fluoro- and chromo-genic materials have been studied for quite some time now, there are still some challenges that need to be addressed. The rising concern for the environment and the use of green chemistry is currently a general trend influencing future research and development. As a result, preparing eco-friendly products through green chemistry is an important milestone. In addition to that, the growing global demand for high-performance colorants combined with the rising consumer preference towards environmentally friendly materials is expected to further drive the global development of dyes and pigments. The ongoing research on novel smart dyes and pigments involving materials of the graphene family is anticipated to have extremely important implications in modern and future technologies by addressing the above contemporary and future challenges. In the future, with the development of novel preparation techniques of graphene and its derivates, it is believed that more and more eco-friendly dyes and pigments based on graphene will be produced, and they will have a great impact on our daily life. Taking all this into account, this review paper has attempted to summarize the most recent achievements in the great technological/scientific research area of graphene-oxide-based fluoro- and chromo-genic materials.

## Figures and Tables

**Figure 1 molecules-27-02018-f001:**
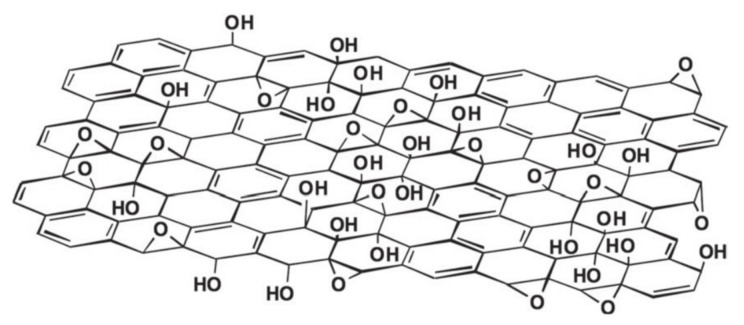
A schematic of graphene oxide structure (Reproduced with permission from ref. [35]. Copyright 2010 Wiley Publications).

**Figure 2 molecules-27-02018-f002:**
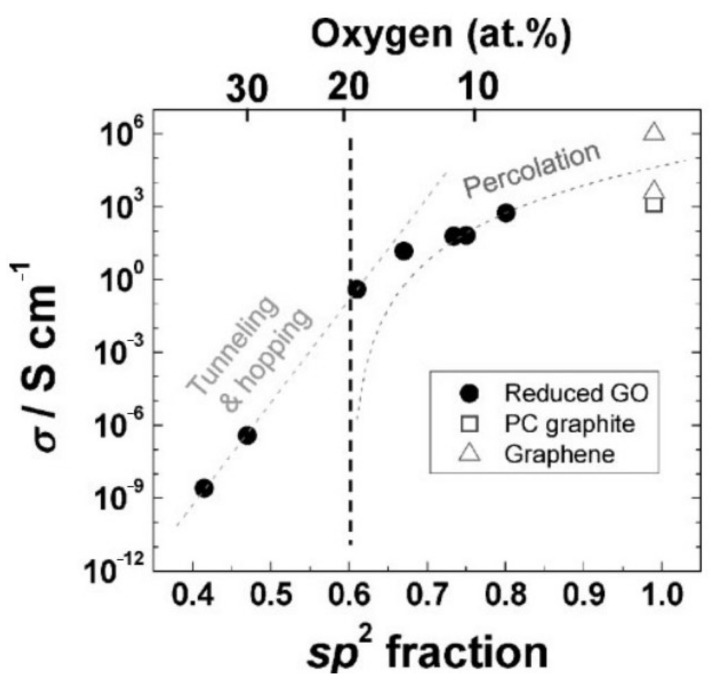
Conductivity of thermally reduced GO as a function of the sp^2^ carbon fraction (Reproduced with permission from ref. [37]. Copyright 2009 Wiley Publications).

**Figure 3 molecules-27-02018-f003:**
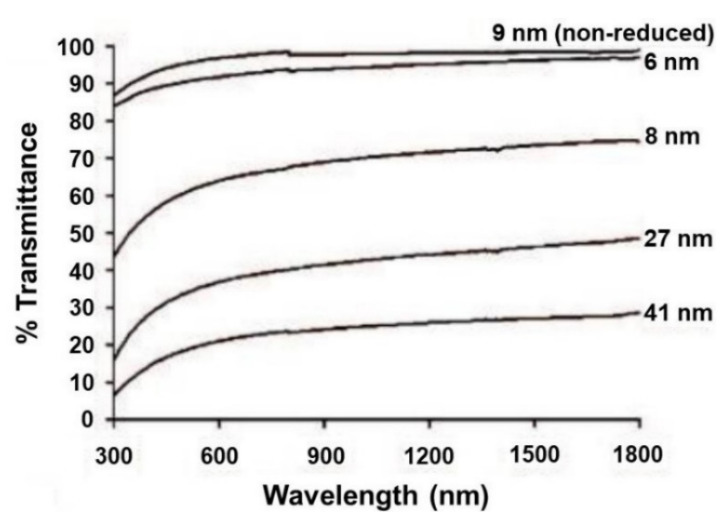
Optical transmittance spectra of the films with indicated thickness (Reproduced with permission from ref. [36]. Copyright 2008 American Chemical Society Publications).

**Figure 4 molecules-27-02018-f004:**
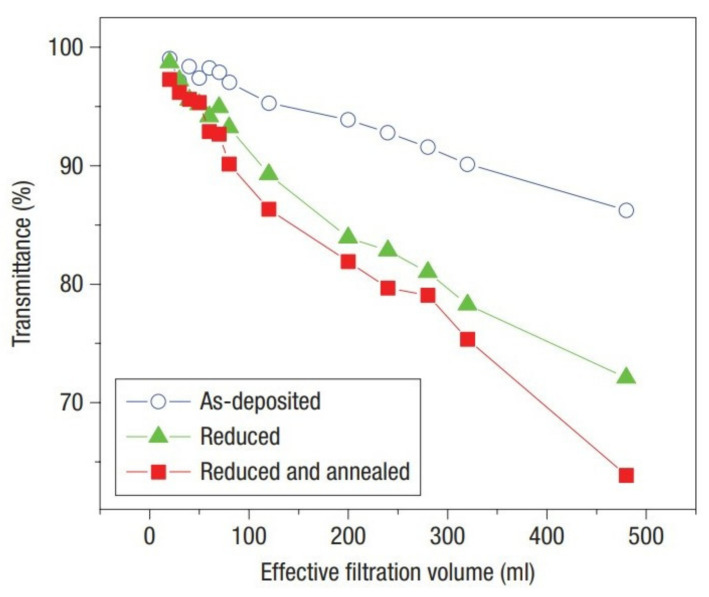
Transmittance at λ = 550 nm as a function of filtration volume for GO thin films prepared by different methods (Reproduced with permission from ref. [41]. Copyright 2008 Springer Nature Publications).

**Figure 5 molecules-27-02018-f005:**
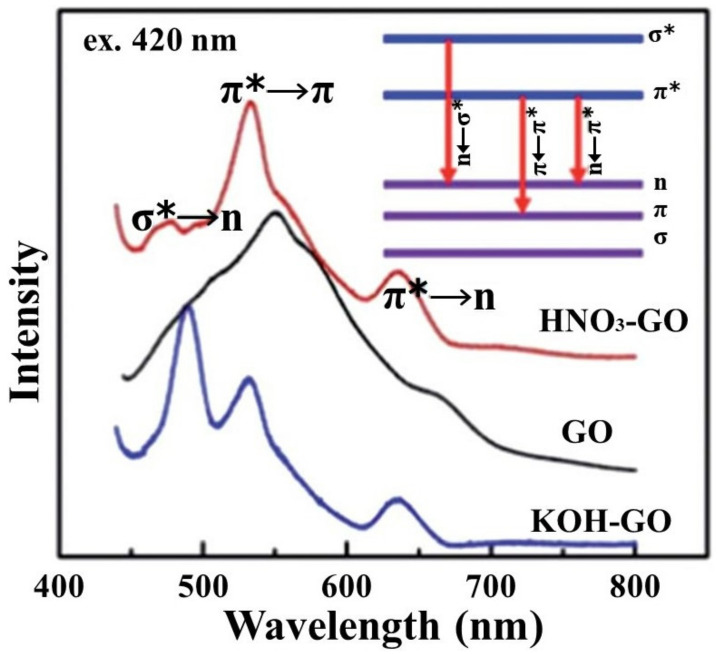
PL spectra of GO and KOH-, and HNO_3_-treated GO sheets under the excitation of 420 nm. Inset: a schematic energy level diagram due to electron transitions (Reproduced with permission from ref. [43]. Copyright 2012 The Royal Society of Chemistry Publications).

**Figure 6 molecules-27-02018-f006:**
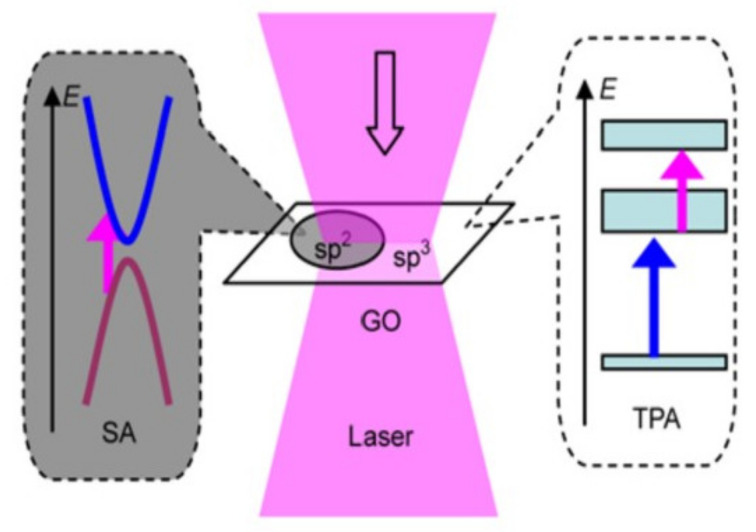
Absorption of light in GO. (Reproduced with permission from ref. [44]. Copyright 2012 Elsevier Publications.)

**Figure 7 molecules-27-02018-f007:**
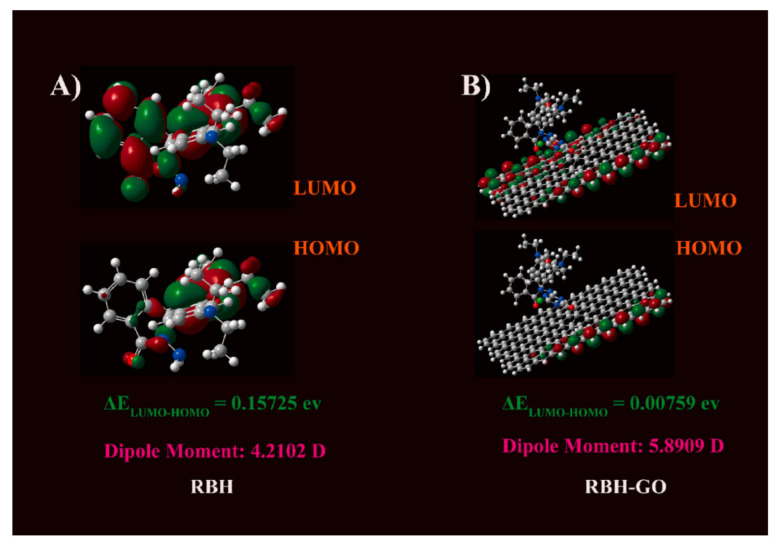
The geometry, energy level, dipole moment, and the electron cloud density distribution of (**A**) RBH and (**B**) RBH-GO. (Reproduced with permission from ref. [58]. Copyright 2021 Elsevier Publications.)

**Figure 8 molecules-27-02018-f008:**
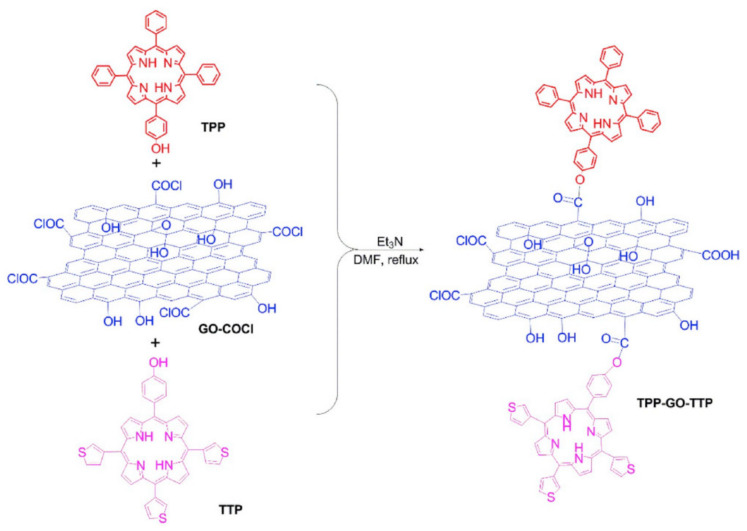
Synthetic route of TPP-GO-TTP (Reproduced with permission from ref. [59]. Copyright 2020 Elsevier Publications).

**Figure 9 molecules-27-02018-f009:**
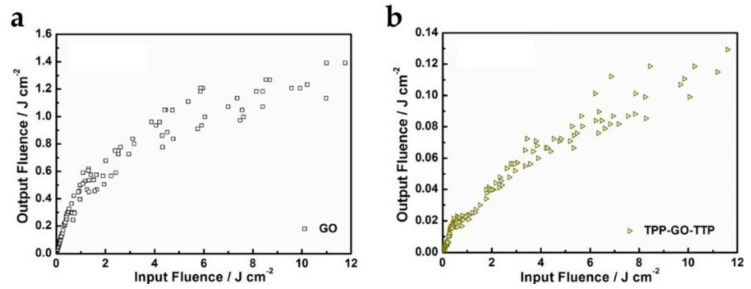
Optical limiting performances of (**a**) GO and (**b**) TPP-GO-TTP (Reproduced with permission from ref. [59]. Copyright 2020 Elsevier Publications).

**Figure 10 molecules-27-02018-f010:**
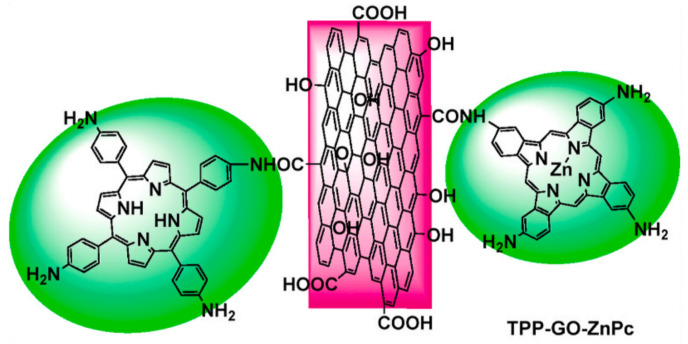
The configuration of TPP-GO-ZnPc (Reproduced with permission from ref. [60]. Copyright 2021 Elsevier Publications).

**Figure 11 molecules-27-02018-f011:**
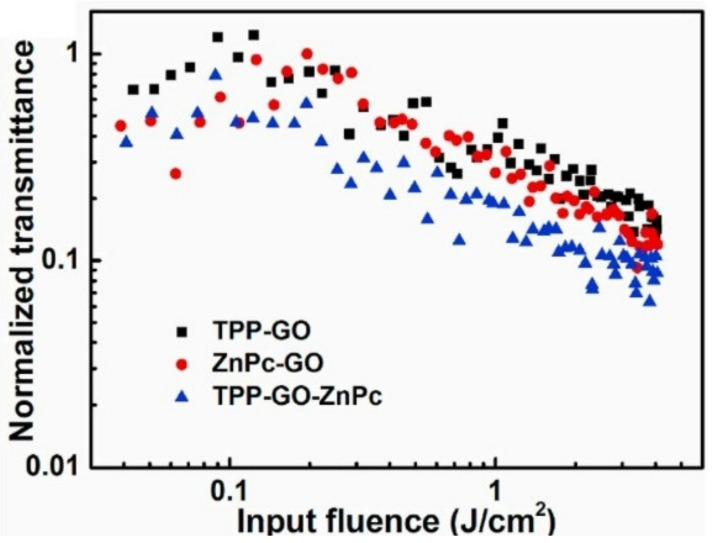
The optical limiting performance of TPP1-GO-ZnPc (Reproduced with permission from ref. [60]. Copyright 2021 Elsevier Publications).

**Figure 12 molecules-27-02018-f012:**
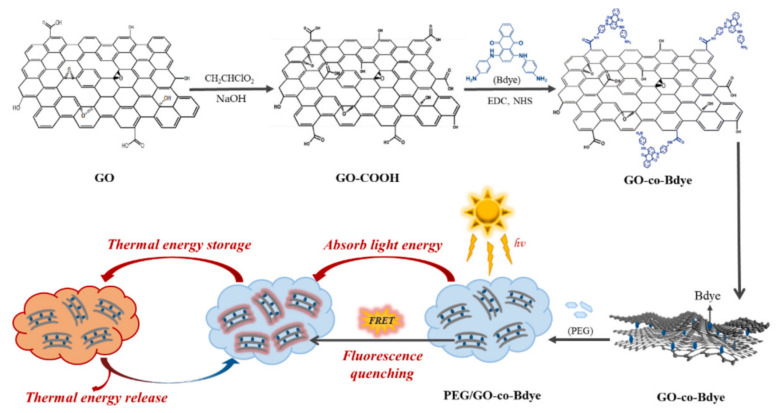
Schematic illustration of the preparation processes of PEG/GO-co-Bdye composite PCMs (Reproduced with permission from ref. [63]. Copyright 2022 Elsevier Publications).

**Figure 13 molecules-27-02018-f013:**
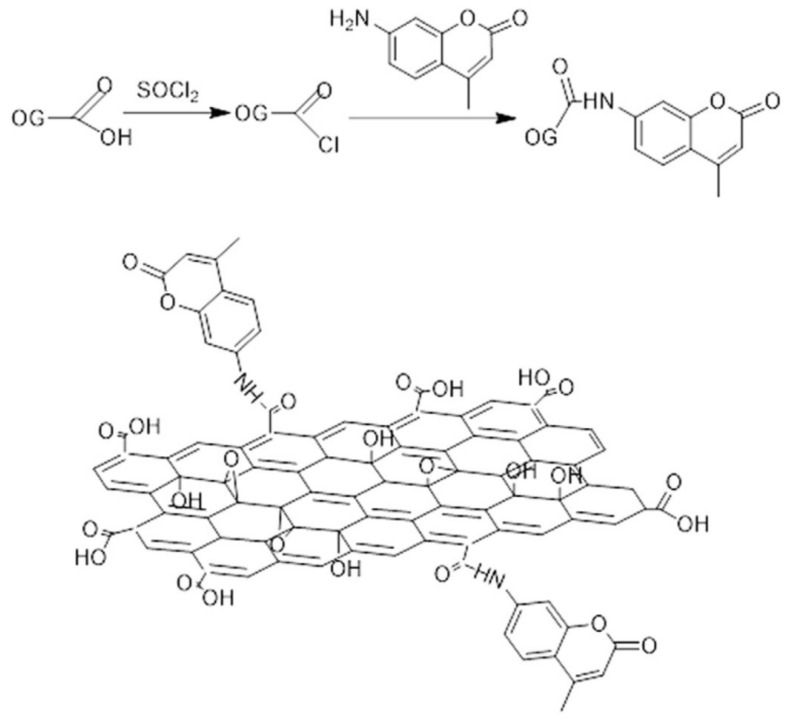
Synthesis of the GO-NH-COUR (Reproduced with permission from ref. [64]. Copyright 2015 Elsevier Publications).

**Figure 14 molecules-27-02018-f014:**
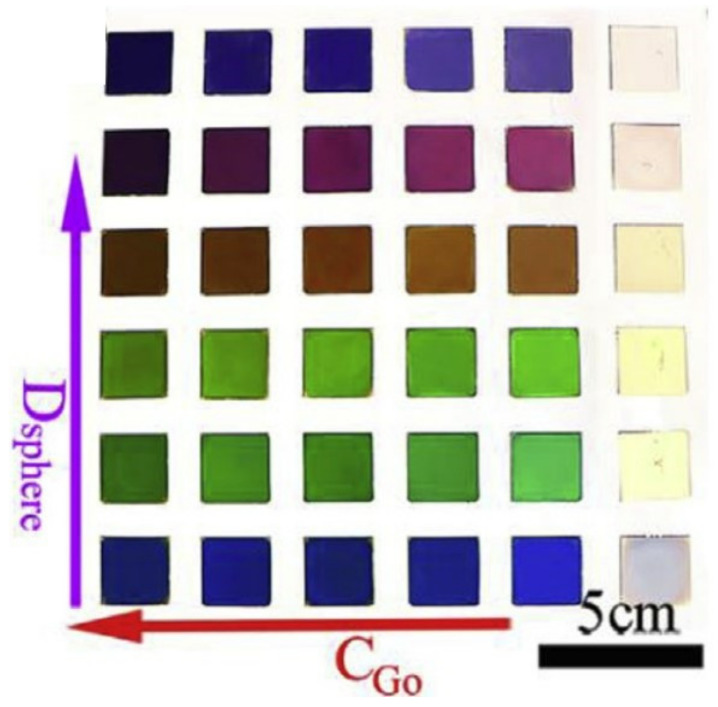
Image of the color cards. The diameter of PS particles gradually increases from bottom to top and the concentration of GO gradually increases from right to left (Reproduced with permission from ref. [83]. Copyright 2019 Elsevier Publications).

**Figure 15 molecules-27-02018-f015:**
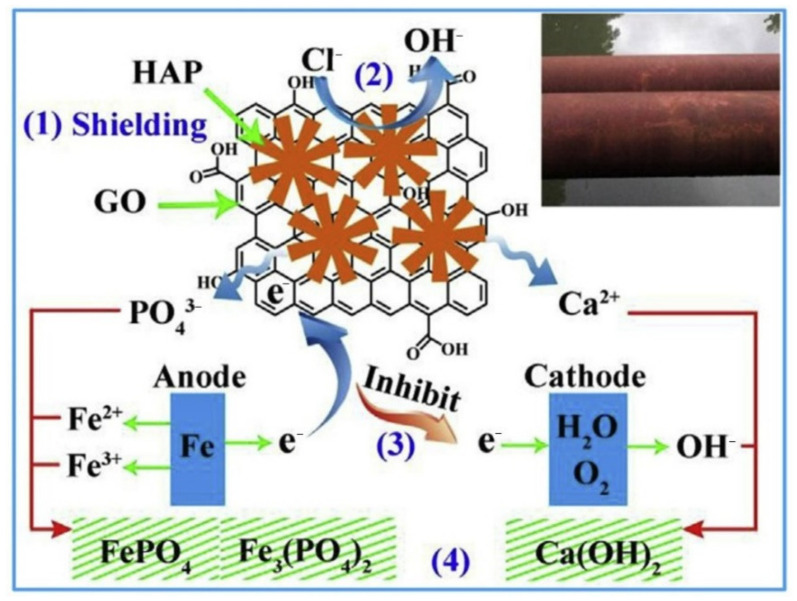
The anticorrosion mechanism of GO-HAP (Reproduced with permission from ref. [84]. Copyright 2019 Elsevier Publications).

**Figure 16 molecules-27-02018-f016:**
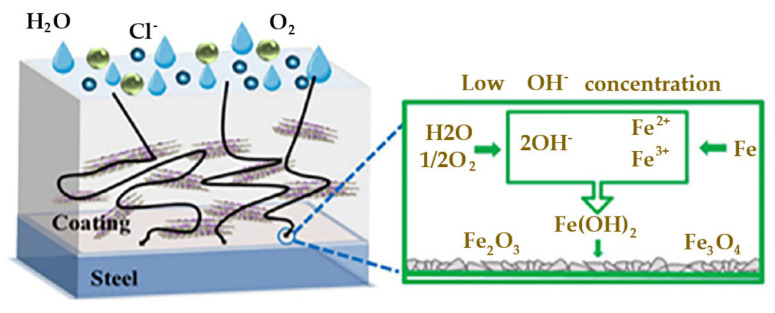
Schematic representation of the anticorrosion mechanism of the coating (Reproduced with permission from ref. [89]. Copyright 2020 Elsevier Publications).

## Data Availability

No new data were created or analyzed in this study. Data sharing is not applicable to this article.

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
