# Peer review of "Graphene-Oxide-Based Fluoro- and Chromo-Genic Materials and Their Applications"

_molecules, 2022, doi:10.3390/molecules27062018_

Round 1

Reviewer 1 Report

GO is currently being used extensively in the fabrication of novel dyes and pigments owing to its excellent properties and special structure. Combining GO and other members of the graphene family with classic dyes and pigments has a great impact on the performances of the newly arising composite materials. This is a very timely review on conjugated polymers for most recent advances in the field of dyes and pigments based on GO as a key ingredient or as an important cofactor. While the important structural variations are well described and the best choices highlighted for their applications, this can help in the design principles very much.

I would like to recommend it for publish in Molecules.

Author Response

We are grateful to Reviewer 1 for this very positive overview on our manuscript. We believe and hope that this work will indeed attract a great deal of interest in various scientific and technology communities dealing with 2D materials and dyes and pigments.

Reviewer 2 Report

In this review, the authors presented the most recent advances in the field of novel dyes and pigments based on GO as a key ingredient or as an important cofactor. The addition of GO has a great effect on the performances of the fabricated dyes and pigments, in terms of the aggregation of dyes, the anticorrosion properties of pigments, the viscosity and rheology of inks, etc, further expanding the range of applications of dyes and pigments. Challenges in the current development as well as the future prospects of GO-based dyes and pigments are also discussed. The review is meaningful, and I would like to recommend its publication after minor revisions

  • For optical limiting devices, the spectra should be given
  • For Anticorrosive Pigments, the thickness of coating layer should be discussed
  • graphene-based inks for supercapacitor should be discussed
  • Pay attention to Journal abbreviation in references
  • References 1-3, page number
  • Reference 7
  • Some related papers might need to be included in the revised manuscript: 10.1186/1556-276X-7-161; 10.1039/C1NR11149C; 10.1002/ADMA.201802403; 10.1002/CEY2.113; 10.1002/CEY2.68
  • Reference 13
  • References 32, 41-43, 46-48, 51, page number
  • References 61-62, 64, 67, 73, 76-78
  • TOC is required

Author Response

Our detailed responses can also be found in the attached docx. file. Reviewer 2 is is kindly informed that the file contains also Figures.

Comments of Reviewer 2

1.For optical limiting devices, the spectra should be given.

Answer: Thanks for your helpful comments. In order to better describe the optical limiting performance of the GO-based dyes, the corresponding spectra have been added in the manuscript (Figure 9 and Figure 11).

Figure 9. Optical limiting performances of (a) GO and (b) TPP-GO-TTP. (Reproduced with permission from ref. [59]. Copyright 2020 Elsevier Publications.).

Figure 11. The optical limiting performance of TPP1-GO-ZnPc. (Reproduced with permission from ref. [60]. Copyright 2021 Elsevier Publications.).

  1. For Anticorrosive Pigments, the thickness of coating layer should be discussed

Answer: Thanks for your helpful suggestion. According to the articles related to epoxy coating doped by GO-based anticorrosive pigments, the coating layer were mostly painted on the steel by wire rod brushing. The thickness of the coating layer was added in the manuscript. And the corresponding section in manuscript has been revised as below.

Additionally, abundant oxygen-containing groups on the surface and edge of GO can also provide potential reactive sites for polymer grafted modification [53, 54, 85-88]. The polymers grafted GO have gained extensive attention in recent years due to their extraordinary advantages. In particularly, they have been widely reported as anticorrosive pigments in waterborne epoxy coatings by forming a layered barrier and increasing the diffusion pathway of corrosive electrolytes. Zhu et al. [89] synthesized a compound pigment of polypyrrole functionalized GO (GO-PPy) by in-situ polymerization method. Incorporation of the nanocomposite into waterborne epoxy coating can obviously improve the corrosion protection performance of coating. The composite coating was painted on the surface of mild steel by wire rod brushing. And the thickness of the coating layer was about 80 μm. The mechanism of corrosion resistance of the GO-PPy coated on mild steel is shown in Figure 16. GO nanosheet perpendicular to the thickness of coating provides a meandering diffusion channel for corrosive ions, which delays the occurrence of corrosion. In addition, the amino function groups of PPy generate a crosslinking effect to the epoxy, decreasing the micro-pores in epoxy. The conducting PPy can accept electrons and accelerate the formation of passivation layer composed of Fe3O4 and Fe2O3.

  1. Graphene-based inks for supercapacitor should be discussed

Answer: Thanks for your advice. The application of the graphene-based inks in supercapacitor  has been added in the manuscript. The corresponding section in manuscript has been revised as below.

Different printing technologies have different requirements for ink viscosity. Reasonable control of ink viscosity is very important to obtain high-performance inks. And they show great potential applications in flexible, thin, and wearable electronics. For example, Lacey et al. [100] printed complex hierarchical porous structures using the extrusion-based printing technology, which were proved as the first 3D printed Li-O2 cathode. The additive-free and printable ink was produced by adding the holey GO (hGO) with high concentration (about 100 mg mL-1) to water. The aqueous hGO ink exhibits shear-shinning behavior, which can be extruded into complex 3D architectures (such as stacked mesh structures) by extrusion-based printing. The printed hGO mesh has multiple levels of porosity from macroscale to nanoscale, which provides pathways for electrolyte and oxygen, improving the performance of Li-O2 batteries. Gonzalez-Dominguez et al. [101] prepared some water-based inks composed of a ternary system (carbon nanotubes, GO, and nanocellulose) by autoclave methods. Importantly, by controlling the experimental conditions, low-viscosity inks, high-viscosity paste or self-standing hydrogels can be obtained. The obtained liquid inks as well as the viscous pastes can be printed into conductive films with low resistivity value (less than 100 Ω/□) by spray coating and rod-coating technologies, respectively. These conductive films possess excellent robustness, which has the ability to avoid decomposition during corrosive treatment with organic solvents, and their electrical properties can be improved through high-temperature treatment, showing great potential in various applications such as batteries and solar cells. Chang et al. [102] prepared a water-soluble graphene@polyvinyl alcohol (PVA)-H3PO4 hybrid ink for the fabrication of microelectrodes of planar supercapacitors by gravure printing technique. The high dispersibility of graphene in PVA-H3PO4 effectively prevents the self-restacking/aggregations of graphene, achieving the enhanced accessibility of electrolyte ions to graphene surface. The flexible supercapacitors shows enhanced electrochemical performances. The increased areal capacitance is 37.5 mF cm-2 at the scan rate of 5 mV s-1. And the maximum energy density of 5.20 μWh cm-2 is obtained at the areal power density of 3.2 mW cm-2. Wang et al. [103] reported a polyaniline/GO (PANI/GO) gel ink for three-dimensional (3D) printing. The addition of GO can effectively adjust the rheological property of PANI, making the viscosity of composite ink meet the requirement of direct ink writing technique. In addition, the reduction of GO can further strengthen the conducting mechanical strength of PANI. The printed PANI/RGO interdigital electrodes of planar supercapacitors exhibits satisfactory electrochemical performances with an areal specific capacitance of 1329 mF cm-2.

Recently, functional graphene-based inks have developed rapidly and they show great potential application in the construction of portable and flexible electronic devices including supercapacitors, sensors, batteries, solar cells, and optoelectronics. On the basis of the requirements of different printing techniques, regulation of the formulation of ink is a crucial subject, because it plays an important role in the performances of printed devices. At present, more and more novel materials with excellent properties are applied into the preparation of composite inks. The exploration of functional inks further promotes the development of electronic devices towards the direction of miniaturization, flexibility and versatility.

  1. Pay attention to Journal abbreviation in references

References 1-3, page number

Reference 7

Some related papers might need to be included in the revised manuscript: 10.1186/1556-276X-7-161; 10.1039/C1NR11149C; 10.1002/ADMA.201802403; 10.1002/CEY2.113; 10.1002/CEY2.68

Reference 13

References 32, 41-43, 46-48, 51, page number

References 61-62, 64, 67, 73, 76-78

Answer: Thanks for your kind advice. We have carefully checked the format of the references and corrected it. In addition, in order to make the content of the manuscript more comprehensive, the related papers (10.1186/1556-276X-7-161; 10.1039/C1NR11149C; 10.1002/ADMA.201802403; 10.1002/CEY2.113; 10.1002/CEY2.68) have been added in the manuscript.

  1. TOC is required

Answer: Thanks for your helpful comments. The TOC has been added in the manuscript.

Reviewer 3 Report

It is widely used in dyes and pigments due to the unique structure and properties of graphene. However, graphene is insoluble in water which greatly limits its application. To solve the issue, in recent years, Scientifics introduced oxygen-containing groups to increase the solubility of graphene in water, which also provide a variety of active sites to facilitate functionalization. In this review, the types of graphene oxide used in dyes and their effects on properties are reviewed. In addition, and the author summarizes the latest achievements of graphene oxide based fluoro- and chromo- genic materials in the field of scientific and technological research by quoting a large number of literatures and outcomes. Based on the reviewer’s suggestions, this paper can be accepted after the minor revision.

  1. The importance of this review should be added in the Abstract section.
  2. In the introduction section, the author described one graph regarding to the dyes and pigments, then transfer to the graphene. The author should give an description the advantage of the graphene compare to the traditional  dyes and pigments.
  3. The authors should make an appropriate summary and overview after each section, which could give a description some significance and inspiration of the reviewed work in detail.
  4. An outlook should be provided.
  5. In the introduction sections, Adv. Sustainable Syst. 2021, 2100244 and https://doi.org/10.31635/ccschem.021.202101483 should be cited as Ref. 4.

Author Response

We would like to kindly inform Reviewer 3 that our detailed responses are also incuded in the attached docx file.

Comments of reviewer 3:

  1. The importance of this review should be added in the Abstract section.

Answer: Thanks for your helpful suggestion. We have made some changes in the Abstract. And the corresponding section in manuscript has been revised as below.

Composite materials and their applications constitute a hot field of research nowadays due to the fact that they comprise a combination of unique properties of each component they consist of. Very often, they exhibit better performance and properties compared to their combined building blocks. Graphene oxide (GO), as the most widely used derivate of graphene, has attracted widespread attention because of its excellent properties. Abundant oxygen-containing functional groups on GO can provide various reactive sites for chemical modification or functionalization of GO, which in turn can be used to develop novel GO-based composites. This review outlines the most recent advances in the field of novel dyes and pigments based on GO as a key ingredient or as an important cofactor. The interaction mechanism of graphene and other materials are highlighted. The special structure and unique properties of GO have a great effect on the performances of fabricated hybrid dyes and pigments including the color performance of dyes, the anticorrosion properties of pigments, the viscosity and rheology of inks, etc, which further expands the applications of dyes and pigments in dyeing, optical elements, solar-thermal energy storage, sensing, coatings, and microelectronics devices. At last, challenges in the current development as well as the future prospects of GO-based dyes and pigments are also discussed. This review provides a reference for the further exploration of novel dyes and pigments.

  1. In the introduction section, the author described one graph regarding to the dyes and pigments, then transfer to the graphene. The author should give a description the advantage of the graphene compare to the traditional dyes and pigments.

Answer: Thanks for your helpful comments. The limitations of the traditional dyes/pigments and the advantages of GO-based dyes/pigments have been added into the manuscript. And the corresponding section in manuscript has been revised as below.

Dyes and pigments play an important role in our daily life because they can be widely used in many applications as key-ingredients in cosmetics, paints, textiles, coatings, plastics, construction materials, food, paper, and printing inks [1]. The global colorants market has in recent years been greatly enhanced as a result of the steadily increasing needs of various end use industries pertaining to dyes and pigments. More and more synthetic dyes and pigments are produced. However, the applications of some synthetic dyes and pigments have detrimental effects on environment [2]. In addition, with the development of science and technology, some traditional dyes and pigments can no longer meet the growing demand for color performance, corrosion resistance of coatings [3], and photoelectric conversion efficiency [4-6] among others. For example, some organic dyes such as phthalocyanines tend to form aggregate due to the unique structures which adversely influences its color performances including color dispersion and strength. Pigments can improve the corrosion resistance of waterborne epoxy resin. Nevertheless, some inorganic anticorrosive pigments have a serious impact on environment. And some traditional pigments-based epoxy can not be used in harsh condition. In order to overcome these limitations and further improve the performance of traditional dyes and pigments, some modification and/or functionalization is necessary. Among the surface modifiers, materials of the graphene family have attracted considerable attention being used as scaffolds or as important cofactors influencing the properties and performance of the composite materials in which they are employed [7].

Graphene is an outstanding representative of the 2D materials family and it is also the most studied 2D material which exhibits a range of remarkable properties such as outstanding mechanical strength, excellent electrical conductivity, high thermal conductivity, good light transmittance, ultrahigh carrier mobility, large surface area and flexibility [8-15]. Furthermore, graphene shows complete impermeability to any gases. These properties make graphene have a very wide range of applications in flexible electronics, protective coatings, solar cells, etc [15-21]. However, graphene is a hydrophobic material and it has no noticeable solubility in molecular solvents, which greatly limits its application. Graphene oxide (GO), as a hydrophilic derivative of graphene, has excellent dispersibility in many solvents, which is ascribed to the presence of large amount of oxygen-containing functional groups on the surface of GO. These functional groups also provide reactive sites for surface modification reactions, thus opening up numerous possibilities for the modification of dyes and pigments. The composite dyes/pigments may show improved performances due to the interaction between functional molecules. For example, Guo et al. [1] prepared GO-modified polymer dyes by introducing GO into aqueous polyurethane based polymer dyes. The modified polymer dyes exhibited high coloration rates and their dry/wet rubbing fastness as well as migration resistance were both improved to grade 5, which promoted their application in high-performance leather dyeing. Sadawy et al. [22] synthesized GO/zinc phosphate composite coating (GO/ZP) on the Al-Zn-Mg alloy substrate by using a chemical conversion coating technique, proving that the addition of GO to ZP bath can improve the corrosion resistance of coating. Due to the special structure of GO, it shows great potential in the fabrication of novel dyes and pigments. In this work, a detailed review of graphene-based dyes and pigments is presented. And the recent progresses of GO-based dyes and pigments, especially the effects of GO on different performances of the synthetic dyes/pigments as well as the applications of GO-based dyes and pigments are highlighted.

  1. The authors should make an appropriate summary and overview after each section, which could give a description some significance and inspiration of the reviewed work in detail.

Answer: Thanks for your kind advice. A brief summary has been added after each section. For example, “Recently, functional graphene-based inks have developed rapidly and they show great potential application in the construction of portable and flexible electronic devices including supercapacitors, sensors, batteries, solar cells, and optoelectronics. On the basis of the requirements of different printing techniques, regulation of the formulation of ink is a crucial subject, because it plays an important role in the performances of printed devices. At present, more and more novel materials with excellent properties are applied into the preparation of composite inks. The exploration of functional inks further promotes the development of electronic devices towards the direction of miniaturization, flexibility and versatility.” The paragraph has been added in section 3.3.2 Graphene-based Inks.

  1. An outlook should be provided.

Answer: Thanks for your advice. And your comments will play an important role in our writing. An outlook of the graphene-based dyes and pigments has been added in section 5. And the corresponding section in manuscript has been revised as below.

GO is currently being used extensively in the fabrication of novel dyes and pigments owing to its excellent properties and special structure. Combining GO and other members of the graphene family with classic dyes and pigments has a great impact on the performances of the newly arising composite materials. For example, GO can effectively regulate the aggregation of dyes, modulate the origin optoelectronic properties of dyes, improve the color strength of pigments, enhance anticorrosion performance of pigments, and lead to the fabrication functional inks. The fabricated composites show remarkable applicability in various technological and scientific areas withing biology and fluorescence imaging, optical elements, solar-thermal energy storage, sensing, coatings, and microelectronics devices. Although GO-based fluoro- and chromo- genic materials have been studied for a long time, there are still some challenges need to be addressed. The rising concern for the environment and the use of green chemistry is currently general trend influencing the future research and development. As a result, preparing eco-friendly products through green chemistry is an important milestone. In addition to that, the growing global demand for high-performance colorants combined to the rising consumer preference towards environmentally friendly materials is expected to further drive the global development of dyes and pigments. The ongoing research on novel smart dyes and pigments involving materials of the graphene family, is anticipated to have extremely important implications for modern and future technologies by addressing the above contemporary and future challenges. In the future, with the development of preparation techniques of graphene and its derivates, it is believed that more and more eco-friendly dyes and pigments based on graphene will be produced. And they will have a great impact on our daily life. Taken all this into account, this review paper attempts to bring up the most recent achievements in the great technological/scientific research area of graphene oxide based fluoro- and chromo- genic materials.

  1. In the introduction sections, Adv. Sustainable Syst. 2021, 2100244 and https://doi.org/10.31635/ccschem.021.202101483 should be cited as Ref. 4.

Answer: Thanks for your kind advice. In order to make the content of the manuscript more comprehensive, the related papers (Adv. Sustainable Syst. 2021, 2100244 and https://doi.org/10.31635/ccschem.021.202101483) have been added in the manuscript.
